# Recent Developments in Molecular Characterization, Bioactivity, and Application of Arabinoxylans from Different Sources

**DOI:** 10.3390/polym15010225

**Published:** 2023-01-01

**Authors:** Jinxin Pang, Yi Zhang, Xiaoyang Tong, Yaoguang Zhong, Fanjun Kong, Dan Li, Xifan Liu, Yongjin Qiao

**Affiliations:** 1Research Center for Agricultural Products Preservation and Processing, Shanghai Academy of Agricultural Sciences, Shanghai 201403, China; 2College of Food Sciences & Technology, Shanghai Ocean University, Shanghai 201306, China; 3School of Health Science and Engineering, University of Shanghai for Science and Technology, Shanghai 200093, China; 4Shanghai Fengxian District Agricultural Technology Promotion Center, Shanghai 201306, China; 5Shanghai Heshou Food Co., Ltd., Shanghai 201812, China

**Keywords:** arabinoxylan, molecular identity, biological activity, applications, products

## Abstract

Arabinoxylan (AX) is a polysaccharide composed of arabinose, xylose, and a small number of other carbohydrates. AX comes from a wide range of sources, and its physicochemical properties and physiological functions are closely related to its molecular characterization, such as branched chains, relative molecular masses, and substituents. In addition, AX also has antioxidant, hypoglycemic, antitumor, and proliferative abilities for intestinal probiotic flora, among other biological activities. AXs of various origins have different molecular characterizations in terms of molecular weight, degree of branching, and structure, with varying structures leading to diverse effects of the biological activity of AX. Therefore, this report describes the physical properties, biological activities, and applications of AX in diverse plants, aiming to provide a theoretical basis for future research on AX as well as provide more options for crop breeding.

## 1. Introduction

AX is a group of hemicelluloses with D-xylose as the building block of the polysaccharide chain; it contains arabinose substituent groups [1]. On the basis of the solubility of AX, AX includes water-extractable arabinoxylan (WEAX) and water-unextractable arabinoxylan (WUAX) [2]. AX from diverse sources varies considerably concerning monosaccharide composition, the molar ratio of arabinose to xylose (Ara/Xyl), structure, and so on. Therefore, various sources can result in different molecular weights and degrees of branching of AX; these structural features affect the biological activity of AX [3].

Recently, attention has been paid to AXs’ potential biological response modifier (BRM) for several years [4]. The unique properties of AX, including ferulic acid content, molecular weight, substitution, and the molar ratio of arabinose to xylose, determine its unique functions in antioxidants, prebiotic properties, immunomodulation, and hypoglycemia. In the aspect of antioxidants, AX shows strong antioxidant activity due to the combination of ferulic acid, coumaric acid, and other phenolic compounds [5]. Malunga et al. [6] found that the antioxidant properties of AX were not only related to the source and total phenolic content but also the degree of xylose residues’ substitution. It can be inferred that the ferulic acid content, average molecular weight, Ara/Xyl ratio, and xylose residue substitution determine the antioxidant capacity of AX. In immunomodulation, AX plays a vital role in the immunomodulatory process of the body. AX was shown to induce dendritic cells to produce the immunomodulatory cytokine Interleukin-10 [7] and strengthen natural killer cell activity [8]. Thus, AX can significantly inhibit the growth of tumor cells. At the same time, Savitha et al. [9] discovered that the immunomodulatory effect of AX was related to its ferulic acid content. In the case of prebiotic properties, AX, which is prebiotic, is responsible for the increase in the number and diversity of intestinal flora [10,11,12]. On the other hand, branched AX could better promote microbial proliferation and improve short-chain fatty acid contents (SCFAs) [13]. Complex trisaccharide side chain structure and terminal xylose content may impact AXs’ prebiotic properties [14]. Due to the fact that the structure-prebiotic properties relationship is complex, the fermentability may be related to Ara/Xyl, which needs further study [15]. In the area of hypoglycemia, AX can promote the growth of intestinal flora, thereby releasing more SCFAs. SCFAs may also stimulate the release of the heterotropic hormone glucagon-like protein 1 (GLP-1) via G protein-coupled receptor 43 (Gpr43) in L cells of gastrointestinal endocrine cells. GLP-1 boosts insulin secretion and improves insulin resistance [16]. Therefore, AX has a hypoglycemic effect because of the increase of GLP-1 in vivo. In addition, several surveys have shown that AX has a variety of biological activities beneficial to humans, as AX helps suppress appetite, delay gastric emptying, and promote bowel movements. In summary, the effect of the AX’s biological activity is related to its structural characteristics. The structure of AX is diverse and complex, and AXs from different sources have various structural characteristics. Therefore, AXs of varying sources have diverse effects on biological activity. Apparently, polysaccharides derived from various plant sources exhibit different properties, which may not be unique to AX. For example, β-glucans are polysaccharides polymerized from D-glucopyranose monomers linked by β-bonds. β-glucan is mainly derived from microorganisms, algae, and cereals. The structural characteristics and biological functions of β-glucans vary due to their diverse origins. In respect to the structure, cereal β-glucan is mainly linked by β-1,3 and β-1,4 glycosidic bonds, while microbial or algal β-glucan is mainly linked by β-1,3 and β-1,6 glycosidic bonds. In terms of biological activity, oat β-glucan can significantly reduce fasting blood sugar, improve glucose tolerance and insulin sensitivity, and achieve a hypoglycemic effect [17]; *Ganoderma lucidum* spores β-glucan and marine diatom β-glucan have immune-enhancing effects [18,19].In terms of application, Lentinula edodes β-glucan, yeast β-glucan and oat β-glucan have various effects on starch pasting, gelation, and digestive properties due to their distinct molecular weights and spatial conformations. [20]. In summary, various sources of polysaccharides may have different effects on bioactivity.

In recent years, AX has been widely used in various fields. First, AX can be used to make flour products, for example, bread, youtiao, and steamed buns [21,22,23]. Pihlajaniemi et al. [24] developed a new extraction process that consists of the pretreatment of the bran by KOH followed by the enzymatic digestion of the bran. The method productively produced an antioxidant bran syrup with high AX content, which was applied to bread making. The bread did not have a significant discrepancy in sweetness and saltiness compared to typical bread, but the bread produced some off-flavors. Therefore, AX can be made into more flavorful bread, for instance, whole wheat bread and sourdough bread. Secondly, AX has also been applied to the field of film. Rye AX films exhibit excellent oxygen barrier properties (between 0.9~1.0 cm^3^ μm/m^2^ d kPa) [25]. In the examples above, AX has been more widely used in flour products and films. With further awareness of the AX’s functional properties, the AX will certainly be more widely used in more areas, bringing more significant social and economic benefits to enterprises.

In overview, more reports have described the relevant aspects of AX. He et al. [26] discussed the extraction method, isolation, purification, and structural characterization of AX. AX has several biological activities, among which the prebiotic effects of AX were described by Chen et al. [27] and Emily Schupfer et al. [28]. Although many reviews focus on extraction, isolation, and purification, these also elaborate on the physicochemical properties and biological activity of AX, and the relationship between the structure and function of AX needs to be explored in depth. However, only a few reviews on the application of AX and related products have been published. Therefore, this paper focuses on the molecular characteristics, biological activities, applications, and related products of AX in different gramineous plants. It aims to provide a theoretical basis for future research on the structure-function relationship of AX and the development of AX-related products while guiding seed source innovation and the breeding of gramineous crops.

## 2. Molecular Characterization of AX from Different Sources

As shown in Table 1, AX is mainly observed in cereals, for example, wheat, rye, oats, highland barley, sorghum, and corn [29]. Then, AX also exists in some dicotyledonous plants, for instance, *Andrographis paniculate* and Lauraeae. AXs from various sources varied in molecular weight, monosaccharide composition, Ara/Xyl, and structure. Table 1 mainly lists the molecular weight, monosaccharide composition, Ara/Xyl, and structural characteristics of AX from various plants, aiming to identify differences in AX from diverse sources.

### 2.1. Molecular Weight

In Table 1, the molecular weight of AX ranges from 126.6–4300 kDa. The molecular weights of some barley AXs and wheat AXs exceed 1000 kDa, and the molecular weights of rye AX and corn AX are around 300 kDa. The molecular weight of sorghum AX is 100–400 kDa, and the molecular weight of dicotyledons AX is around 100 kDa; their molecular weights are smaller than those of gramineous plants. Some wheat and barley AXs have the highest molecular weights, probably because the hydrogen bonding in this AX causes the molecules to aggregate [30]. In addition, Hromádková et al. [31] also revealed that AX can form aggregates through carbohydrate-protein binding or through the cross-linking of ferulic acid substituents, resulting in larger molecular weights.

### 2.2. Monosaccharide Composition

Monosaccharide composition and Ara/Xyl are described as measures of the degree of AX branching. As can be seen in Table 1, the AXs of most plants consist of arabinose, xylose, glucose, and galactose. Among them, mannose is found in hull-less barley bran AX, wheat flour AX, rye grain AX, sorghum seed AX, and sorghum bran AX; rhamnose is present in wheat bran AX, sorghum AX, lacquered sorghum bran AX; and there is glucuronic acid in all of them. On the other hand, xylose accounts for the largest proportion of AX, followed by arabinose, with galactose and glucose accounting for a relatively small proportion. However, barley hull AX and some varieties of rye grain AX contain 20% glucose, which may contain some glucan, resulting in a high glucose content [32].

The lower the Ara/Xyl, the smaller the degree of AX branching. The pattern and extent of arabinose substitution vary with plant origin and plant tissue location [33]. Within Table 1, Ara/Xyl ranges from 0.11–2.85. The Ara/Xyl of sorghum bran AX, the Ara/Xyl of sorghum seeds AX, and the Ara/Xyl of Lauraeae AX are more than 1, and the Ara/Xyl of wheat AX is around 0.9, demonstrating a high degree of branching. On the contrary, the Ara/Xyl of oats AX, the Ara/Xyl of barley hull AX, the Ara/Xyl of corn stover AX, and the Ara/Xyl of *Andrographis paniculate* AX are below 0.5, indicating a lower degree of branching.

### 2.3. Structural Feature

The xylose in AX is connected by β-1,4 glycosidic bonds to form xylan, which constitutes the backbone of AX. α-L-furan arabinose may be monosubstituted or disubstituted at the O-2 and O-3 positions. Moreover, the substitution group may contain ferulic acid, 4-O-methylglucuronide [34]. The structural features of diverse sources AX mainly lie in the differences in the type and content of the substituent groups or the substitution positions. In Table 1, 14.78% O-3 is monosubstituted and 10.76% O-2,3 is disubstituted in the AX backbone in hull-less barley bran; 22.1% O-2/3 is monosubstituted and 18.4% O-2,3 is disubstituted in the AX backbone in peeled barley kernels; in the xylan backbone of rye bran AX, it contains 57.71% unsubstituted pyranose residues and 6.22% disubstituted xylose. Secondly, the structural features of pangola grass also confirm its high degree of branching, with 60% of it consisting of substituent groups. This means that different AXs have all kinds of degrees of substitution, and the structural differences are mainly reflected by the various substitution positions and degrees of substitution of arabinose.

**Table 1 polymers-15-00225-t001:** Molecular characterization of AX from different sources.

	AX Source	Extraction Method	Molecular Weight/kDa	Monosaccharide Composition (mol%)	Ara/Xyl	Structural Feature	Reference
Barley	Hulless barley bran	0.375 mol/L NaOH	298.36	Ara:Xyl:Gal:Glc: Man = 30.13:51.55:10.33:5.09:2.90	0.58	14.78% O-3 is monosubstituted, 10.76% O-2,3 is substituted	[35]
	Barley hulls	1 mol/L NaOH of 5% NaBH_4_	4300	Ara:Xyl: Glc = 13.1:55.9:28.3	0.23	/	[36]
	Peeled barley seeds	1% NaBH_4_ in saturated Ba(OH)_2_	1360	Ara:Xyl: Glc = 30.3:48.5:2.7	0.60	Monosubstituted (O-2/3, 22.1%) and doubly substituted (O-2,3, 18.4%)	[37]
Wheat	Wheat flour	0.26 mol/L NaBH_4_ in saturated Ba(OH)_2_	2000	Ara:Xyl:Gal:Glc: Man = 46.6:48.6:1.3:2.5:1.0	0.96	β-The xylan backbone is present on the (1→4) bond and is substituted at O-3 or O-2 and O-3.	[38]
	Wheat bran	0.44 mol/LNaOH	/	Ara:Xyl:Gal:Glc:Man: Rha = 27.8:29.7:2.0:2.9:0.1:0.1	0.94	/	[39]
Rye	Rye grain	Hot water extraction	156	Ara:Xyl:Glc = 24.5:52.9:1.3:	0.46	/	[32]
	Rye grain	NaOH extraction	309	Ara:Xyl:Gal:Glc: Man = 20.9:44.4:0.3:20.3:1.8	0.47	/	
	Rye bran	1% (*w*/*v*) NaBH_4_ in saturated Ba(OH)_2_	380	Ara:Xyl: Glc: = 36.53:61.31:2.16	0.60	The xylan skeleton contained 57.71% unsubstituted xylan residues and 6.22% disubstituted xylose.	[40]
Sorghum	Sorghum seeds	1% (*w*/*v*) NaBH_4_ in saturated Ca(OH)_2_	223.9	Ara:Xyl:Gal:Glc:Man: Rha = 47.53:43.82:2.34:4.81:1.11:0.39	1.09	The polysaccharide backbone is 1,4-β- D -xylan, which is replaced by α- l -arabinose residues mainly at the O-2 or O-3 sites	[41]
	Sorghum bran		126.6	Ara:Xyl:Gal:Glc:Man: Rha = 49.37:45.45:0.13:2.74:1.78:0.52	1.09	The polysaccharide backbone is 1,4-β- D-xylan, which is mainly substituted by α- l -arabinose residues at the O-2 or O-3 sites	
	Lacquer sorghum bran	Alkali extraction	363	Ara:Xyl:Gal:Glc:Rha:GalA: GlcA = 34.60:48.85:3.07:8.13:0.98:1.00;3.37	0.71	/	[42]
Corn	Corn Bran	Alkali extraction	362	Ara:Xyl:Gal:Glc:Rha:GalA: GlcA = 27.46:48.52:12.08:4.28:0.43:1.02:6.21	0.56	/	[42]
	Corn stover		367	Ara:Xyl:Gal:Glc:Rha:GalA: GlcA= 18.14:52.69:10.94:9.66:1.33:1.77:5.47	0.34	/	
Oats	Oat grain	0.26 mol/L NaBH_4_ of saturated Ba(OH)_2_	6–2000	Ara:Xyl:Gal:Glc: UA = 27:43:3:2:4	0.43	/	[43]
	Oat grain	0.26 mol/L NaBH_4_ of 6 mol/L NaOH	100	Ara:Xyl:Gal:Glc: UA = 9:78:1:10:2	0.11	/	
Dicotyledonous plants	Pangola grass	Hydroborates	/	Ara:Xyl:Gal: Glc = 32.9:42.8:20.7:3.7	0.77	Highly branched arabinoxylan, 60% xylose double substitution	[44]
	*Andrographis paniculate*	4% NaOH	149	Ara: Xyl = 20:80	0.25	Constructed as the 1,4-alpha-D-xylose backbone	[45]
	Lauraeae	Water lifting	175	Ara:Xyl = 74:26	2.85	Constructed as the 1,4-alpha-D-xylose backbone, substituted with furan arabinose at C2	[46]

Note: “/” indicates that something is not mentioned in the literature. Ara is arabinose; Xyl is xylose; Gal is galactose; Glc is glucose; Rha is rhamnose; Man is mannose; GalA is galacturonic acid; GlcA is glucuronide; UA is glyoxylate.

## 3. Biological Activity of AX from Various Origin Sources

Both gramineae and some dicotyledons contain AX. Due to the diversity of the AX structure, the functions of AX are also diverse. Therefore, AX derived from diverse plants may have various biological activities.

### 3.1. AX from Wheat

Wheat is a cereal crop widely grown worldwide and is one of the staple foods of human beings. Wheat is ground into flour to make bread, buns, cookies, noodles, and other foods, while after fermentation, it can be made into beer, alcohol, liquor, and so on. In Table 2, wheat grain and wheat bran contain AX. Diverse investigations have shown that wheat AX has different beneficial effects on the human body, such as prebiotics and immune-enhancing effects. First of all, on the prebiotics side, Figure 1a lists the beneficial effects of AX on the human body in terms of its prebiotic properties. AX can enhance the growth of probiotics and inhibit the growth of non-probiotics. Candela Paesani et al. [47] explored the effect on prebiotic properties through Argentine soft wheat AX and durum wheat AX. They identified a significant promotion effect of AX on the growth of *Bifidobacterium*. Durum wheat AX was more effective in inhibiting *Bifidobacterium* growth, and both wheat AXs inhibited *Clostridium perfringens*. Thus, it confirmed the effectiveness of AX as a potential prebiotic. The degree of substitution of arabinose or ferulic acid was negatively correlated with fermentability [48]. The Ara/Xyl of durum wheat AX in the above study was smaller than that of soft wheat AX, but the fermentation rate of durum wheat AX was higher than that of soft wheat AX, and this phenomenon exactly verified the conclusion. In the same way, Candela Paesani et al. [49] evaluated the in vitro fermentation of AX from two types of wheat and came to conclusions consistent with the studies mentioned above. AX improved the number of *Bifidobacterium* and *Lactobacillus* and decreased pH; it also increased short-chain fatty acid content. The results demonstrated that AX could enhance probiotic activity.

In immunomodulation activity and antitumor activity, Moerings et al. [50] first demonstrated that wheat-derived arabinoxylan preparations induce training and resilience in human macrophages. This experiment suggested that AX could promote the secretion of interleukins and tumor-damaging factors by macrophages (Figure 1b), thereby enhancing the immune function and antitumor ability of the body. According to the study, they added Argentine soft wheat AX and durum wheat AX to the culture medium of human colon cancer cells-116 (HCT-116) and determined the cell viability after 24h, 48h, and 72 h of incubation, respectively. Compared with the control group, the cell viability of the experimental groups decreased. This result indicated that AX could inhibit the growth of cancer cells, and the study also demonstrated that AX enhanced the cell viability of macrophages and splenocytes [51]. The biological activities of the two wheat AXs exhibit varying effects, resulting from the differences in glycosidic bonds and arabinose substitutions.

### 3.2. AX from Barley

Barley is one of the world’s oldest cultivated crops, with a variety of uses such as food, forage, brewing, medicinal, and so on. Barley is also rich in AX, which has a beneficial effect on the human body. More and more researchers have found that barley AX has better hypoglycemic and immunomodulatory abilities. To begin with, Kento et al. [16] extracted AX from barley flour and experimentally studied the hypoglycemic ability of AX in mice. They discovered for the first time that AX promotes the secretion of GLP-1, which in turn promotes the secretion of insulin to decrease blood glucose. Furthermore, AX can increase the SCFA content owing to its fermentation characteristics. The improvement in SCFA can affect L-cell differentiation and GLP-1 secretion genes, thus enabling an even better release of GLP-1.

Secondly, among the immunomodulatory effects, barley leaves AX were able to promote the secretion of immunoglobulin A (IgA). It was concluded that they promote the increased secretion of IgA by inducing the secretion of recombinant human transforming growth factor (TGF-b1), granulocyte monocyte colony-stimulating factor (GMCSF), and interleukin-6 (IL-6) [52]. These cytokines are secreted by various lymphocytes associated with IgA production, mainly T helper cells [53]. This phenomenon further suggested that the substance has immune-enhancing effects. One additional note: they found an abundant class of immunoglobulins in the intestine, secretory IgA. IgA is considered to be the first line of defense against harmful substances and toxins in the intestinal epithelium [54]. Therefore, an increase in IgA content can demonstrate an increase in immune function.

### 3.3. AX from Sorghum

Sorghum is distributed mainly in tropical, subtropical, and temperate regions of the world. Sorghum is processed and called sorghum rice for consumption or made into noodles, pancakes, rolls, and other flour foods. Few studies have been conducted on the biological activity of sorghum AX, and only a few have investigated the antioxidant activity of sorghum AX. Fabiola E [55] proposed the antioxidants white sorghum AX, red sorghum AX, and high-tannin sorghum bran AX. Furthermore, it showed that these three types of sorghum AXs had high antioxidant capacity, while high-tannin sorghum bran AX exhibited the highest antioxidant activity. The explanation for this is that AX contained a large number of tannins with antioxidant activity. 

### 3.4. AX from Rice

Rice is among the most important food crops in the world today, and more than half of the global population uses rice as their primary food source. Rice bran is the outer layer of rice, which can be used to extract oil and make pickles. Equally, rice bran contains a considerable amount of AX. Several studies have proved that rice bran AX has antitumor ability and antioxidant capacity. On the one hand, Nariman K. et al. [56] demonstrated the chemopreventive effect of rice bran AX on the development of hepatocellular carcinoma in mice. They made mice carcinogenic by N-nitrosodiethylamine (NDEA) and CCl_4_, and fed mice biological bran containing AX. The results indicated that the AX-containing biobran could inhibit hepatocarcinogenesis by inducing apoptosis and inhibiting inflammation and cancer cell proliferation. Meanwhile, it has been reported that rice bran AX can also destroy neuroma cells through in vivo and in vitro experiments [8], which is more robust evidence that rice bran AX has antitumor ability.

On the other hand, rice bran AX had a higher chemical antioxidant capacity with a higher DPPH scavenging rate and Fe^2+^ reduction capacity, mainly because of the higher content of this AX-bound phenolic acid. The biological antioxidant capacity of this AX in cells was also high but not related to the content of bound phenolic acids [57].

### 3.5. AX from Corn

Corn is native to Central and South America, then widely cultivated in tropical and temperate regions around the world. China is one of the countries with the largest cultivation area. With the massive production of corn processing by-products, it is essential to achieve resource recycling to reduce resource waste. Corn cobs, corn stover, and corn bran contain substantial amounts of AX, and AX in corn bran can promote intestinal probiotics’ growth. Xu et al. [58] further studied the in vitro fermentation characteristics of corn bran AX. The experimental studies revealed that AX could promote gas production and SCFA production by intestinal flora, which suggested that AX could promote the growth of intestinal flora. This study illustrates that the high branching complexity of corn bran AX causes a gradual slowing of the fermentation characteristics of the intestinal flora, consistent with the conclusions drawn from wheat AX prebiotics. Meanwhile, Hubert W. Lopez et al. [59] have also demonstrated that corn bran AX promotes the proliferation of intestinal flora through in vivo experiments in rats; the study also demonstrated the ability of AX to lower plasma cholesterol levels in rats. 

Besides, AX also has an antioxidant capacity, and some studies showed that AX with a higher hydroxycinnamic acid content had a stronger antioxidant capacity [60].

### 3.6. AX from Other Plants

*Andrographis paniculata* AX has been shown to have antioxidant activity. The EC_50_ of Fe^2+^ chelating activity, the EC_50_ of superoxide radical scavenging, and the EC_50_ of hydroxyl radical scavenging of *Andrographis paniculata* AX are less than 500 μg/mL and have potent antioxidant activity [45].

**Table 2 polymers-15-00225-t002:** Biological Activity of AX from Different Sources.

	AX Source	Bioactivity	Experimental Subjects/Experimental Method	Biochemical Parameters	Results	Reference
Wheat	Argentine soft wheat, Argentine durum wheat	Prebiotics	C57BL/6 male rat	Intestinal flora, SCFA	↑Intestinal flora, ↑SCFA	[47]
	Argentine soft wheat, Argentine durum wheat	Prebiotics	In vitro fermentation	pH, Gas production pressure, SCFA, *Bifidobacteria*, *Lactobacillus* abundance	↓pH, ↑Gas production pressure, ↑SCFA, ↑*Bifidobacteria*, *Lactobacillus* abundance	[49]
	Wheat bran	Immunomodulation	Human monocytes	IL-6, TNF-α	↑IL-6, ↑TNF-α	[50]
	Argentine soft wheat, Argentine durum wheat	Antitumor	HCT-116 colon cancer cells, macrophages, splenocytes	Cellular Viability	↓Cellular Viability	[51]
Barley	Barley flour	Hypoglycemic, prebiotic	C57BL/6J male mice	GLP-1, SCFA, Cecal chyme intestinal flora	↑GLP-1, ↑SCFA, ↑Cecal chyme intestinal flora	[16]
	Barley leaf	Immunomodulation	C3H/HeN rats	IgA; TGF-b1, GMCSF, IL-6	↑IgA; ↑TGF-b1, ↑GMCSF, ↑IL-6	[52]
Sorghum	Sorghum bran	Antioxidant activity	In vitro antioxidant assay	ORAC	High antioxidant capacity	[55]
Rice	Rice bran	Antitumor	Male Wistar rats	p53, Bax, Bcl-2, caspase-3, NF-κB/p65	↑p53, ↑Bax, ↓Bcl-2, ↑caspase-3, ↓NF-κB/p65	[56]
	Rice bran	Antitumor	Peripheral blood mononuclear cells; Mice	Cytotoxicity, TNF-α, IL- 6, IL-8,	↑Cytotoxicity, ↑TNF-α, ↑IL- 6, ↑IL-8,	[8]
	Skimmed rice bran	Antioxidant activity	HepG2 cells (cellular antioxidant assay)	DPPH clearance rate, Fe^2+^ reducing ability, ROS	Stronger DPPH clearance and Fe^2+^ reduction	[57]
Corn	Corn Bran	Prebiotics	In vitro fecal fermentation	Gas production, SCFA	↑Gas production, ↑SCFA	[58]
	Corn Bran	Prebiotics	Male Wistar rats	Appendix quality, SCFA, pH	↑Appendix quality, ↑SCFA, ↓pH	[59]
	Corn fiber	Antioxidant activity	In vitro antioxidant assay	ORAC	Stronger resistance to oxidation	[60]
Other Plants	*Andrographis paniculata*	Antioxidant activity	In vitro antioxidant assay	Fe^2+^ chelating ability, superoxide radical scavenging rate, hydroxyl radical scavenging rate	Stronger resistance to oxidation	[45]

Note: SCFA is short-chain fatty acid; IL-6 is interleukin-6; TNF-α is tumor destruction factor alpha; GLP-1 is glucagon-like protein 1; TGF-b1 is recombinant human transforming growth factor; IgA is an immunoglobulin; GMCSF is granulocyte monocyte colony-stimulating factor; ORAC is oxygen radical absorbance capacity; IL-8 is interleukin-8; p53 is an oncogene; Bax is a monoclonal antibody. Blc-2 is the B lymphocytoma-2 gene; Capase-3 is cysteinyl aspartate specific proteinase; and NF-κB/p65 is the nuclear factor-κB subunit p65 affinity peptide. ROS is reactive oxygen species; ↑ indicates an increased level/enhanced effect; ↓ indicates a decreased level/reduced effect.

## 4. Applications

China has a vast consumer market for cereals and a large production of cereals every year, generating a large number of cereal processing by-products. The by-product contains a large amount of AX, and the AX has biological activities such as hypoglycemia, antioxidant activity, immune enhancement, and so on. Therefore, to achieve the maximum utilization of resources, AX can be obtained from cereal processing by-products and applied to products. Consequently, the application of AX has a broad development prospect.

### 4.1. Flour Products

AX has a high viscosity and can be used in the dough to enhance the viscoelasticity of the dough, which helps optimize the process of making dough products. AX significantly raised the storage modulus of the dough, indicating that the dough is more elastic, demonstrating that AX can be applied to dough products [61]. AX is divided into WEAX and WUAX. WEAX is helpful for the dough to form small gluten agglomerates, improves the stability of the gluten protein network, and makes the dough form a uniform and dense gluten network [62]. Moreover, it can increase the strength and elasticity of the gluten in the dough to obtain a more resilient dough [63]. WUAX is added to the dough to protect the gluten protein’s hydrogen and disulfide bond conformations [64]. However, WUAX can contribute to a large and uneven hole in the product, which eventually causes the flour product to harden [65]. After adding WEAX and WUAX to the youtiao, the results demonstrated that WEAX and WUAX changed the conformation of gluten molecules and inhibited the formation of some disulfide bonds, as well as the thermal aggregation of gluten proteins. WEAX reduced gluten’s surface hydrophobicity and increased gluten’s solubility, while WUAX showed the opposite results [22]. WUAX and WEAX exhibit different properties in the dough. Therefore, the properties of both AXs need to be further examined to develop the flour products better.

### 4.2. Wine

Most beers are fermented with wheat malt. One study revealed that 30% rye flour was used to replace some malts for fermentation. Furthermore, the results demonstrated that adding rye flour significantly increased the viscosity of the beer, because a large amount of AX was released, and AX itself had a better viscosity during the fermentation process [66].

### 4.3. Films

AX has excellent film-forming properties. Recent experiments in this area suggest that sorghum bran AX is sensitive to moisture absorption under high relative humidity conditions. It has fine strength properties and has the potential to be a renewable material [67]. Corn AX are used in the field of film. The maximum tensile degree of the produced corn AX film is 29.3 MPa, and the tear resistance is 0.3 N. The film has a high tensile degree and strong tear resistance, and it can be applied to food packaging materials [68]. Similarly, another publication showed that Young’s modulus of films made from corn cob AX increased to 1400–1600 MPa and the strength improved to about 53 MPa [69], making this AX a suitable matrix material for film applications. These are enough to prove that AX can make high-quality films. What is more, AX has also been studied in edible coating films. AX can also apply a coating film to fruits and vegetables to preserve and extend their storage period. Usman Ali et al. [70] applied AX with the β-glucan stearic acid ester (SABG) to preserve apples and confirmed that the application of an AX-SABG coating could prolong the shelf life of apples.

### 4.4. Other Fields

AX is also used in stabilized emulsions. LV et al. [71] enzymatically synthesized wheat bran-AX-bovine serum albumin conjugate using peroxidase and H_2_O_2_ as catalysts. The combination improves the stability of the emulsion and enables it to withstand changes in environmental conditions. This method optimizes the emulsification properties of the emulsion and provides a better option for processed products that require emulsified food products. In Table 3, shows the application of AX. 

## 5. AX Related Products

AX has a variety of biological activities. It is often added to food and health products as a functional factor. Additionally, as mentioned above, AX can be added to films to enhance the stretch of the film. Thus, AX film products are promising. There are 166 AX-related products patented in the data of the State Intellectual Property Office, some of which are shown in Table 4. In the Amazon Mall (www.amazon.com), AX products have health products and dog food, such as Biobran MGN-3 and Cheddar Cheese.

### 5.1. Food

According to research, AX can increase the dough’s tensile viscosity and has better rheological properties when applied to flour [72]. For example, patent No. CN111317093A describes a special flour for steamed buns. The unique powder for steamed buns contains ≥3.6% of AX. The steamed buns made with this special steamed bun flour are characterized by rich nutrition, excellent dough processing performance, and good product flavor and taste.

Secondly, AX has superior gel characteristics, and the hydrogel formed by AX has a honeycomb-like three-dimensional mesh structure [73]. Based on the superior gelation properties, people made a fish ball with improved gelation properties (Table 4).

Finally, AX has a hypoglycemic function. Therefore, a functional instant tea with hypoglycemic activity was prepared from wheat AX, barley AX, and barley AX, which has a strong flavor and excellent taste.

### 5.2. Film Products

The films prepared by AX have good tear resistance and elasticity [68], which laid the foundation for the development of film products. AX-based film products are mainly environmentally friendly food packaging film, ferulic acid-arabinoxylan copolymer antibacterial film, a biodegradable frozen food packaging film with photothermal antibacterial function, and other products (Table 4).

### 5.3. Other Products

In addition, AX is applied to other products. AX is used as a raw material or additive to make various products, such as compounding emulsion thickener, preservative for shrimp dodgers, and transparent conductive paper for direct printing.

## 6. Conclusions and Prospects

AX has won widespread attention from researchers because it can have health functions on the human body or alleviate human diseases, including promoting the growth of intestinal probiotic flora, antitumor activity, immunomodulation, hypoglycemia, and more. However, AX comes from a wide range of sources, its structure is diverse, and there are discrepancies in the biological activities of diverse structural AX. In order to be able to study the structure-function relationship of AX clearly, this paper reviews the molecular characterization and biological activities of different sources of AX to lay the theoretical foundation for the study of the structure-function relationship. Meanwhile, AX-related products have been developed and launched into the consumer market, where they are well accepted by consumers. Therefore, this paper also reviews and discusses the application of AX and related products, aiming to provide new perspectives for future research.

The biological activity of AX is closely related to its structure, such that different sources of AX will exhibit various biological activities. However, the structure-bioactivity relationship of AX still needs to be clarified. Therefore, more in vivo and in vitro experiments are needed to clarify the specific structure of AX and achieve the desired biological function.

The natural extract AX has various physiological functions and can be used as an active ingredient in dietary supplements, functional foods, pharmaceuticals, etc. AX is used in food, chemical, and bioengineering applications. Both WEAX and WUAX cause foods to exhibit different properties and sensory characteristics, and further research is needed to investigate the role played by both types of AXs in foods. As a renewable hemicellulose, AX has promising cross-linking properties and can be used for the preparation of gels, films, etc. The development of AX products needs to be further explored.

## Figures and Tables

**Figure 1 polymers-15-00225-f001:**
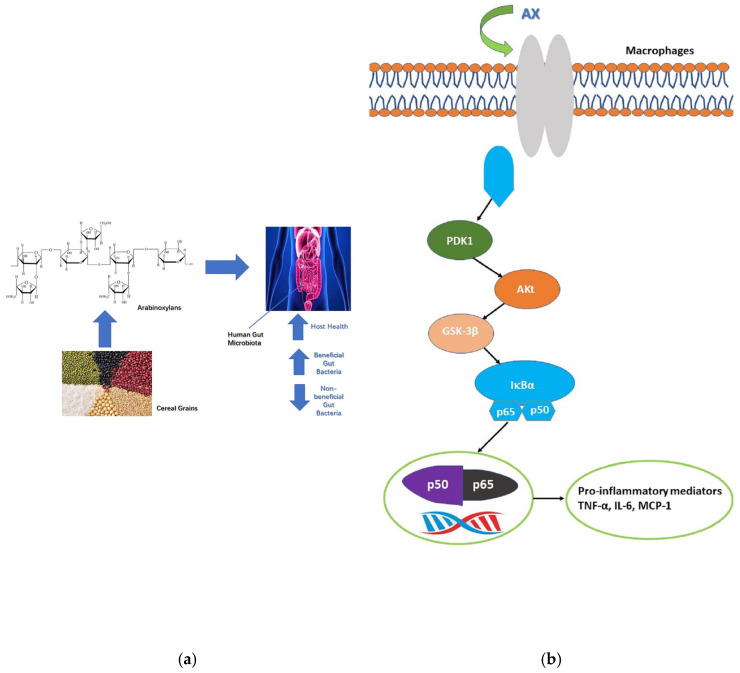
Bioactivity of AX. (**a**) prebiotic properties; (**b**) immunomodulatory activity of AX.

**Table 3 polymers-15-00225-t003:** Application of AX.

Application	Main Component	Effect	Reference
Dough	WEAX, WUAX	WEAX improves the stability of the gluten protein network and makes dough form a uniform and dense gluten network. WUAX can protect gluten protein’s hydrogen and disulfide bond conformation	[62,63,64,65]
Youtiao	Wheat AX	Inhibits the formation of partial disulfide bonds and inhibits the thermal aggregation of gluten proteins	[22]
Beer	Rye flour, wheat malt	Improving beer viscosity	[66]
Film	Corn bran AX	Maximum tensile strength 29.3 MPa, tear resistance 0.3 N	[68]
Film	Corn cob AX	Young’s modulus increased to 1400–1600 MPa and strength increased to about 53 MPa	[69]
Fruit and vegetable composite coating film	AX, SABG	Extended storage period	[70]
O/W Emulsion	Wheat bran AX, bovine serum albumin	Enhanced stability, improved optimization performance	[71]

**Table 4 polymers-15-00225-t004:** AX product patents.

	Patent Number	Name of Product	Main Component	Feature
Food	CN111317093A	Flour for steamed buns	Wheat dextrin layer powder	Nutrient-rich, excellent dough processing performance, good product flavor and texture
	CN112841568A	Fish balls with improved frost resistance and nutritional value	Surimi protein, starch, AX	High water-holding capacity, good frost resistance, high gel characteristics, low digestibility of starch
	CN110897023A	Wheat dextrin layer arabinoxylan hypoglycemic instant tea	Wheat AX, Barley AX, Green Barley AX, Hops, Orange Peel, Pu’er Tea Extract	Good taste, with a strong flavor, lowering blood sugar, safe and no side effects, suitable for a wide range of recipients
Film Products	CN114058055A	Biodegradable frozen food packaging film with photothermal antibacterial function	Wheat bran AX	Easy to prepare, low cost, green and safe, photothermal thawing
	CN114773689A	Ferulic acid-arabinoxylan copolymer antibacterial film	Ferulic acid-arabinoxylan copolymer	Good barrier performance to water vapor and oxygen, good antibacterial properties
	CN114015104A	Environmentally friendly food packaging film	AX	Anti-icing and anti-fouling accelerated freezing, biodegradable
Other Products	CN113647609A	Compounding emulsifying thickener	Konjac gum, xanthan gum, sodium carboxymethyl cellulose, WUAX	Good water retention, adhesion, foam stability, and oxidation resistance
	CN111955537A	Preservatives for shrimp dodgers	Rice bran protein-AX complex, lignan, arbutin	Inhibits the production of spoilage bacteria, and volatile salt nitrogen (TVB-N), and prevents the oxidation of unsaturated fatty acids, etc.
	CN112037962A	Direct printable transparent conductive paper	AX	Simple operation and high reproducibility

## Data Availability

This article is a review article, so there is no data to share.

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
