# Peer review of "Recent Developments in Molecular Characterization, Bioactivity, and Application of Arabinoxylans from Different Sources"

_polymers, 2023, doi:10.3390/polym15010225_

Round 1
Reviewer 1 Report
This paper reports on the “RECENT DEVELOPMENTS in MOLECULAR CHARACTERIZATION, BIOACTIVITY, and APPLICATION of ARABINOXYLANS from DIFFERENT SOURCES”. Manuscript seems be corrected.
I have few comments to the manuscript:
1. Remove “Caps Lock” in title.
2. All manuscript. Deleted extra space.
3. All manuscript. Put a space between words and square brackets “[27]also” to “[27] also”.
4. All manuscript. Explain the abbreviations used in the text when using them for the first time.
Taking into account all comments the manuscript may be published in Polymers after minor revision.
Reviewer 2 Report
The manuscript by Pang et al. surveys research articles published on arabinoxylans. The authors have provided a comparative analysis amongst arabinoxylans from different sources. The authors have put efforts to collect information about various properties of arabinoxylans extracted from different plant materials. Generally, the article is well organized, and contents are enough to warrant it as a review article. Few changes can improve this article.
As it is obvious that components extracted from different plant sources exhibit different properties and this may not be a unique property to arabinoxylan. Can author comment on this statement and bring some examples to strengthen their hypothesis that provided the base for this review? This point should be emphasized in the introduction.
L16: Do not stress it as ‘non-starch’. It is not linked with starch
L20 Write ‘..ability to proliferate intestinal probiotic flora…’
L36: Modify
L43: Replace ‘concluded’ with ‘inferred’
L56-57: It is not clear whether authors are referring to ‘alleviation’ of hypoglycemic effect or its cause.
L79: Check if it is probiotic or prebiotic
L90: Why authors are emphasizing over being ‘non-starch’. There are many non-starch polysaccharides. AX is also ‘non-cellulosic’. It is better to avoid this phrase
It is difficult to differentiate between AX and A/X with the symbols. It would be better to refer A/X with its full form
Italicize names of microorganisms
L168: Write ‘…activity’
Fig. 1: Do not repeat words in legends
L195: It is with reference to a study describing ‘recombinant’ protein. Rewrite to emphasize over it. Instead of ‘increase of IgA’ write secretion of Ig A or increased secretion of IgA.
L261: Bring meaning to this sentence
L269-270. Write full forms
L294: Avoid repetition of the words
Round 2
Reviewer 2 Report
Authors have incorporated most of the changes. Just minor corrections and an overall, but mdoerate level language editing is required.
Still some names are not italicized. Even at some places written with small initial letter such as in the line 174. Check throughout the paper
Table 1, structural properties of barley, write in numbers
